# Proposals for Surmounting Sensor Noises

**DOI:** 10.3390/s23063169

**Published:** 2023-03-16

**Authors:** Andre Pittella, Timothy Sands

**Affiliations:** 1Sibley School of Mechanical and Aerospace Engineering, Cornell University, Ithaca, NY 14850, USA; 2Department of Mechanical Engineering (SCPD), Stanford University, Stanford, CA 94305, USA

**Keywords:** sensor fusion, sensor noise, optimization, feedback, real-time optimization, velocity-based controller

## Abstract

Classical and optimal control architectures for motion mechanics in the presence of noisy sensors use different algorithms and calculations to perform and control any number of physical demands, to varying degrees of accuracy and precision in regards to the system meeting the desired end state. To circumvent the deleterious effects of noisy sensors, a variety of control architectures are suggested, and their performances are tested for the purpose of comparison through the means of a Monte Carlo simulation that simulates how different parameters might vary under noise, representing real-world imperfect sensors. We find that improvements in one figure of merit often come at a cost in the performance in the others, especially depending on the presence of noise in the system sensors. If sensor noise is negligible, open-loop optimal control performs the best. However, in the overpowering presence of sensor noise, using a control law inversion patching filter performs as the best replacement, but has significant computational strain. The control law inversion filter produces state mean accuracy matching mathematically optimal results while reducing deviation by 36%. Meanwhile, rate sensor issues were more strongly ameliorated with 500% improved mean and 30% improved deviation. Inverting the patching filter is innovative but consequently understudied and lacks well-known equations to use for tuning gains. Therefore, such a patching filter has the additional drawback of having to be tuned through trial and error.

## 1. Introduction

Artemis I will be the first integrated flight test of NASA’s deep space exploration system: the Orion spacecraft, Space Launch System (SLS) rocket, and the ground systems at Kennedy Space Center in Cape Canaveral, Florida. The first in a series of increasingly complex missions, Artemis I (Figure 1) will be an uncrewed flight that will provide a foundation for human deep space exploration and demonstrate our commitment and capability to extend human existence to the Moon and beyond. During this flight, the uncrewed Orion spacecraft will launch on the most powerful rocket in the world and travel thousands of miles beyond the Moon, farther than any spacecraft built for humans has ever flown, over the course of about a three-week mission [1].

### 1.1. Introduction to the Problem

Dealing with the fusion of potentially poor, noisy sensors is a ubiquitous challenge that has a long lineage leading to several disparate approaches. Figure 2 illustrates a use case, where noise from position and velocity sensors degrades the performance of state-of-the-art nonlinear adaptive spacecraft control methods. Similar to the problem defined in Section 2, the simulated machinery is performing an angular reorientation.

### 1.2. Review of the State-of-the-Art Alternatives to Address the Problem

Some researchers focus on improving the sensor itself or its signal using internal algorithms. Others seek to develop architectures that prove robust to fused noisy sensor data. Wang et al. proposed a novel method for accurate, autonomous and real-time orbit determinations for geo-referencing with a standalone global positioning system receiver [5]. Xiong et al. addressed systematic centroid errors and poor attitude accuracy by augmenting star trackers with an image intensifier [6]. Kim et al. proposed an algorithm for determining the orbit of a geostationary satellite using single-epoch measurements from a global positioning system receiver with sparse visibility of the global positioning system satellites [7]. Takayama illustrated weaknesses with global navigation system signals and proposed novel sensor noise models used to enhance sensor sensitivity [8].

Leake et al. proposed dealing with sensor issues with improved sensor algorithms, proposing a non-dimensional star identification algorithm compared in terms of accuracy, speed, and robustness to the so-called pyramid algorithm [9]. Marin et al. sought to enhance star trackers by sensor and sensor fusion algorithms to provide a smoother and faster output [10]. Perov et al. sought to utilize the principle of phase interferometer, using multiple receiving antennas [11]. Wang et al. sought to integrate communications navigation with global positioning system sensors [12]. Christian proposed autonomous augmentation using optical navigation by relativistic perturbation of starlight [13]. Fan et al. investigated a plume noise suppression algorithm based on star point shape and the angular distance between stars [14]. Opromolla et al. proposed dealing with sensor issues algorithmically by using a model-based three-dimensional template matching technique for pose acquisition of an uncooperative space object [15].

Rather than dealing with sensor issue algorithmically as proposed by Opromolla, Chen et al. instead focused on the control algorithm using the sensor measurements and proposed a velocity-based impedance control scheme illustrating efficacy in both speed and robustness [16]. This manuscript parallels the insights of Chen et al. and enhances the velocity-based logic with optimization methods of Pontryagin akin Sandberg’s recent recitation [17]. Sandberg’s very recent improvements follow a lineage of small improvements from its provenance in the nonlinear adaptive control methods of the 1990s, as proposed by Slotine [18], and afterward improved by Fossen [19], Sands [20,21], who offered experimental validation in [22]. Most recently, Raigoza [23] augmented the method with autonomous collision avoidance, and Wilt [24] evaluated efficacy in the face of uniform variations in mass moment of inertia (e.g., from fuel slosh). This manuscript proposes and illustrates methods that assume velocity-based control logic like Chen but seek to induce open-loop optimal results like Sandberg, Raigoza, and Wilt, and thus mitigate deleterious effects of noisy sensors.

Controllers are differential equations that are designed to be able to manipulate some state variable by arbitrarily changing a variable connected to it through a physical law or equation. They often feature several points of adjustment and tunability to fit figures of merit desired by the designer. The figures of merit utilized here include spacecraft tracking accuracy (how well the controller reaches the desired target), precision (as measured by deviation), and cost (numerical evaluation of the resources used). Computational burden is also monitored since difficulties such as singular matrix inversions are involved. A variety of control system architectures were evaluated, and their ability is assessed to perform the task of a rest-to-rest reorientation normalized to unity. The different architecture’s performance under the previously mentioned figures of merit will be compared to show how they respond to a same noise environment, and which one best mitigates the effects of noisy sensors.

### 1.3. Novelties Presented

Open-loop optimal results are analytically calculated providing a performance benchmark for comparing other methods.Inspired by Opromolla, 2015 [11], velocity-based classical control is investigated, especially since it was utilized as the comparative benchmark by Sandberg (2022) [17], Raigoza (2022) [23], and Wilt (2022) [24], while the open-loop optimal results in item #1 are used as comparative benchmark here.Real-time optimal control is compared since it was the latest cited proposals of 2022, and can be considered state of the art for this application, as an analytic optimal solution is found, which is not always possible for a generic system.Double-integrator patching filters are implemented seeking to match open-loop optimal results amidst the fusion of noisy sensors.System-inverting patching filters are implemented, also seeking to match open-loop optimal results amidst the fusion of noisy sensors.

### 1.4. Feedback Control System Topology 

This paper will focus on changes to the control architecture (depicted in Figure 3) rather than making direct improvements or modifications to the sensors. Noise is accepted as a fact of the environment, and the main effort is to test control architectures on how they are able to perform in the presence of said noise. The desired state and system dynamics are defined in the following section.

## 2. Control System Architectures

This section offers formulas used in the study, and the implementation of the formulas in simulations is provided in Appendix A and Appendix B. The aim of each architecture, and the selection of equations thereof, is to reach the desired state outlined in Section 2.1, even in the presence of noise from the sensor. The formulas provided show how the control of different architectures is computed, but not explicitly how they work around noise.

### 2.1. The Task at Hand

The controllers will perform a rest-to-rest reorientation of one unit of rotation (indicated by variable θ whose rate is indicated by variable ω) scaled to unity over unit of time scaled to unity. The equations for such a maneuver are listed in Equations (1)–(3). Controller cost will be used as a key figure of merit, where a quadratic cost computation will be used indicated by variable *J* in Equation (4).
(1)θ(0)=ω(0)=0
(2)$$\theta \left(1\right)=\theta_{d}=1$$
(3)ω(1)=0
(4)$$J=\frac{1}{2}\int_{0}^{1}u{\left(t\right)}^{2}dt$$
with state θ(t), rate ω(t), desired state $\theta_{d}$, quadratic cost functional J(t), and control variable u(t). The state, rate, and control are connected in accordance with Equations (5) and (6).
(5)$$\omega =\dot{\theta }$$
(6)$$u=\frac{1}{I}\dot{\omega }$$
for moment of inertia I, where dotted variables indicate derivatives in time.

### 2.2. Proportional Plus Velocity (P+V) Control

The P+V controller is a form of classical control, tuned using two different gains: a proportional gain $K_{P}$ applied to the state error, added to a velocity gain $K_{V}$ applied to the rate, *not the rate error* (see (14)). P+V controllers can be easily tuned to fit desired settling times and damping ratios. For a maximum tolerance of 2% error at the desired settling time, Equation (7):(7)$$t_{s}=-\frac{\ln\left(0.02\times \sqrt{1-\zeta^{2}}\right)}{\zeta \omega_{n}}$$
can be used, where ζ is the damping ratio of the controller and $\omega_{n}$ is its natural frequency response. For a P+V controller, these are known to be Equations (8) and (9).
(8)$$\omega_{n}=\sqrt{K_{P}}$$
and
(9)$$\zeta =\frac{K_{V}}{2\sqrt{K_{P}}}$$
For a settling time of ≤1 s and a damping ratio of 0.7, the gains are calculated to be $K_{P}=37$ and $K_{V}=8.5$.

P+V controllers are always able to reach the desired value, but it is the selection of gains that affects how quickly it is reached. This ability is hampered by noise, especially on the rate sensor. Because the end rate is 0, any error in rate detection will cause a control to move the system off of the desired value. Additionally, the settling time is only for the state target, so no estimation of the rate target settling time is used to set the gains.

### 2.3. Open-Loop Optimal Control

For a given problem with dynamics, boundary conditions, and a cost functional, an optimal solution exists that minimizes the output of the cost functional, which is found using Pontryagin’s method [25]. Using Equations (1)–(6), the task described in *the task at hand* (from Section 2.1) has an optimal control $u^{*}$ of the form in Equation (10).
(10)$$u^{*}\left(t\right)=\theta_{d}\left(at+b\right)$$

By integrating twice and applying the initial and final boundary conditions, the linear system in Equation (11) is found:(11)$$\left[\begin{array}{cccc} 0 & 0 & 0 & 1\\ 0 & 0 & 1 & 0\\ 1/6 & 1/2 & 1 & 1\\ 1/2 & 1 & 1 & 0 \end{array}\right]\left\{\begin{array}{c} a\\ b\\ c\\ d \end{array}\right\}=\left\{\begin{array}{c} 0\\ 0\\ \theta_{d}\\ 0 \end{array}\right\}$$
where the solution to Equation (11) can be applied to (10) to yield Equation (12): (12)$$u^{*}\left(t\right)=\theta_{d}\left(-12t+6\right)$$
which has a minimal quadratic cost of $J^{*}=6$ and exact achievement of the target end conditions. While open-loop optimal control yields the perfect results on paper, it is completely blind to noise and perturbation.

### 2.4. Real-Time Optimal Control

Real-time optimal control, or RTOC, is a modification of the previous iteration of optimal control that considers the current state of the system, allowing the controller to adjust for noise and perturbations. The top two rows of the matrix in (11) are the equations for $\theta^{*}\left(t\right)$ and $\omega^{*}\left(t\right)$, the optimal forms of the state and rate variables, evaluated at the initial conditions. By modifying these rows such that the forms are evaluated at the current time $t_{0}$, the linear system changes to:(13)$$\left[\begin{array}{cccc} t_{0}^{3}/6 & t_{0}^{2}/2 & t_{0} & 1\\ t_{0}^{2}/2 & t_{0} & 1 & 0\\ 1/6 & 1/2 & 1 & 1\\ 1/2 & 1 & 1 & 0 \end{array}\right]\left\{\begin{array}{c} a\\ b\\ c\\ d \end{array}\right\}=\left\{\begin{array}{c} \theta \left(t_{0}\right)\\ \omega \left(t_{0}\right)\\ \theta_{d}\\ 0 \end{array}\right\}$$

The vector of unknowns on the left-hand side is calculated by inverting the matrix and multiplying it by the vector of knowns on the right-hand side. The unknowns a and b are then used in (10) to make an instantaneous optimal control, calculated for the current state of the system. Therefore, the controller can adjust its control output in the presence of noise and perturbations.

While being able to change course based on the current state of the system is an advantage over open-loop optimal control, RTOC has a large computational demand in its inversion of a 4 × 4 matrix. Additionally, the matrix becomes singular as $t_{0}\to 1$, which causes problems as singular matrices are uninvertible. In simulation, the effect of this is the control shooting off to infinity towards the end of the runtime, leading to huge errors in the state and/or rate. To avoid this behavior, a switch is worked in so that, when the matrix determinant approaches 0, the controller switches to open-loop optimal control for the remainder of the runtime.

### 2.5. Patching Filter: Double Integrator

One way to combine the analytical accuracy of optimal control and the computational simplicity and tunability of classical P+V control is to impose a ‘patching’ filter that integrates the optimal control $u^{*}\left(t\right)$ twice to find the optimal state function $\theta^{*}\left(t\right)$, which is given to a P+V controller as the desired value instead of a constant command $\theta_{d}$.

### 2.6. Patching Filter: Double Integrator, Tuned

However, the result of the previous architecture may not be desirable. Should the gains of the original P+V controller be modifiable, tuning them around the patching filter is a viable strategy. This architecture is not as well studied as the original P+V control, so handy equations based around desired settling time and damping ratio are not available. Therefore, the gains will be tuned by hand, i.e., trial and error, until the result meets acceptable figures of merit.

### 2.7. Patching Filter: Control Law Inversion

Should the gains not be modifiable, there is another way to adjust the architecture. Instead of directly integrating the optimal control, the control law for the P+V controller can be solved for an optimal input such that, after all the gains are applied and the control calculated, the P+V controller output matches that of the optimal control $u^{*}\left(t\right)$, where the asterisk ‘*’ indicates optimality. The P+V control law is listed in Equation (14). Solving for the desired state results in Equation (15).
(14)$$u=K_{P}\left(\theta_{d}-\theta \right)-K_{V}\omega$$
(15)$$\theta_{d}=\frac{1}{K_{P}}u+\frac{K_{V}}{K_{P}}\omega +\theta$$

Entering the optimal values $u^{*},\omega^{*},$ and $\theta^{*}$ gives the optimal input. For a patching filter that works in the Laplace domain as a transfer function, this can be rewritten as Equation (16), which can be simplified to become Equation (17).
(16)$$\Theta_{d}\left(s\right)=U^{*}\left(s\right)\left(\frac{1}{K_{P}}+\frac{K_{V}}{K_{P}s}+\frac{1}{s^{2}}\right)$$
(17)$$\frac{\Theta_{d}\left(s\right)}{U^{*}\left(s\right)}=\left(\frac{s^{2}+K_{V}s+K_{P}}{K_{P}s^{2}}\right)$$

Equation (17) is the patching filter to be applied to the optimal control yielding the optimal state trajectory input for the P+V controller.

## 3. Results and Analysis

The following simulations were performed in MATLAB Simulink, using the ode4 (Runge—Kutta) integrator and a 0.01s step size. Using a random number generator with a Gaussian distribution, a noise signal was added to the state and rate with variances of 0.0001 with zero mean. The moment of inertia was also varied randomly but uniformly to ±10% of its true value.

### Monte Carlo Simulation

Displayed in Figure 4 are the results of 1000 simulations of each of the control architectures outlined in the previous section. Table 1 contains the numerical performances of each architecture for direct comparison.

From Table 1, the relationships between accuracy (the final values of θ & ω), precision (the magnitude of their spread, seen in σ), and cost (*J*) can be evaluated. The P+V controller, while having the smallest standard deviations for both values, has notable error in $\omega_{f}$ and the highest quadratic cost, 301% the cost of the next-highest cost. Open-loop optimal has some of the most accurate mean values, but the second-highest spread in both state and rate. Depicted in Figure 5 and Figure 6, RTOC is similar, with lower cost (−0.5%) and rate spread (−29%) but higher state spread (38%) and final rate error (218%). 

The double-integrator patching filter has the second-lowest spreads in both state and rate and the lowest cost, but the final values are completely off the mark; 16% error in state and 943% more rate error than the next highest architecture. After some tuning, it performs similarly to the P+V controller but with higher spreads (17.2% and 264%, respectively), lower cost (−66.8%), and a more accurate final rate (12% the error of the P+V). Finally, the control law inversion patching filter strikes a balance between all figures of merit: lowest state error, third lowest state spread, second lowest rate error, third lowest state spread, and a cost only marginally higher (0.7%) than the optimal $J^{*}=6$, where the asterisk ‘*’ indicates optimality. However, it took the longest to run after RTOC.

## 4. Conclusions

From the results shown in Section 3 summarized in Table 2, there is no one control that performs the best in every single regard. 

### 4.1. Recapped Research

Velocity-based classical control was investigated in the prequels and established as the comparative benchmark here where open-loop optimal results in item #2 were used for initial comparison. *The controller achieved the best angle tracking deviation, but the worst cost performance.*Open-loop optimal results were presented analytically providing a performance benchmark for comparing other proposed methods.Real-time optimal control was presented and compared to the benchmarks.Double-integrator patching filters were introduced seeking to match open-loop optimal results amidst fusion of noisy sensors. *These filters achieved the best velocity tracking performance.*System-inverting patching filters were implemented also seeking to match open-loop optimal results amidst fusion of noisy sensors. These patching filters achieved the best overall performance with notable improvements amidst no dramatic performance degradation in any categories of performance.

### 4.2. Concluding Results

The P+V controller has high cost and rate error, the optimal controls have low cost and high accuracy but higher susceptibility to sensor noise, while the double-integrator patching filter is difficult to tune and costly. The control law inversion patching filter, however, can be seen as the best alternative to open-loop optimal control, especially in the presence of sensor noise and inertia uncertainties, as it had a cost only slightly higher (0.7%) with smaller standard deviations (35.6% less spread in state, 29.5% less spread in rate) and final results more accurate than any other architecture (14.3% being the state error of the runner-up, 18.4% the rate error of the runner-up). The key tradeoff was computational burden since control inversion optimization took 25–40% longer to run than some of the other architectures.

**Recommendation:** *In instances where performance must be induced on pre-existing systems, control inversion patching filters are advised.* □

## Figures and Tables

**Figure 1 sensors-23-03169-f001:**
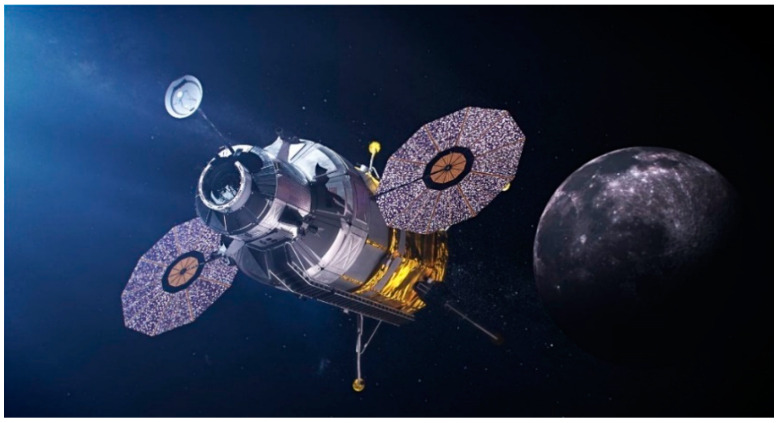
Lunar lander concept sought for the Artemis program, image courtesy NASA [2], image use in accordance with NASA image use policy. Reprinted/adapted with permission from Ref. [3]. NASA.

**Figure 2 sensors-23-03169-f002:**
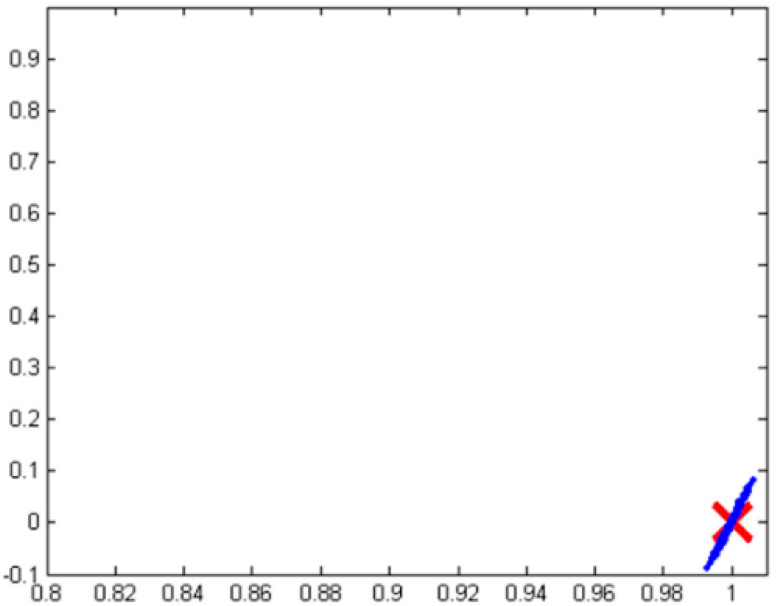
Illustration of deleterious effects of position and rate sensor noise on tracking performance of modern algorithms. Image taken from [4]. The desired state is indicated by a red ‘x’, and the result of simulations are the blue data points. Horizontal axis is angular position and vertical is angular velocity (rate), similar to the state being used in the problem defined in Section 2.

**Figure 3 sensors-23-03169-f003:**
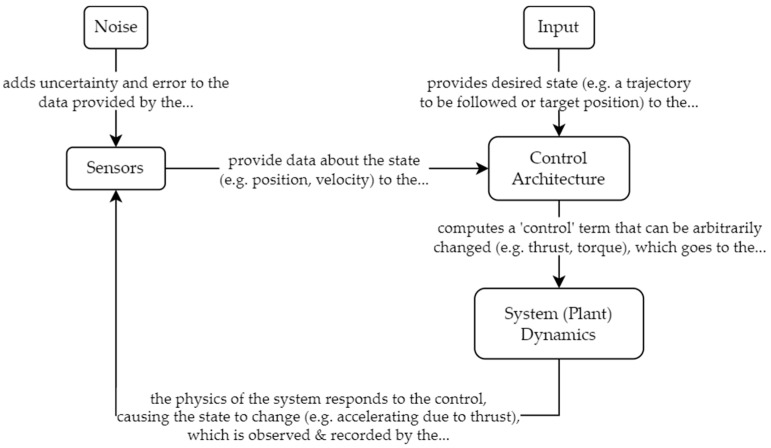
Flowchart showing the topology of a generic system, explaining the connection between the sensors, the control architecture, and the system dynamics.

**Figure 4 sensors-23-03169-f004:**
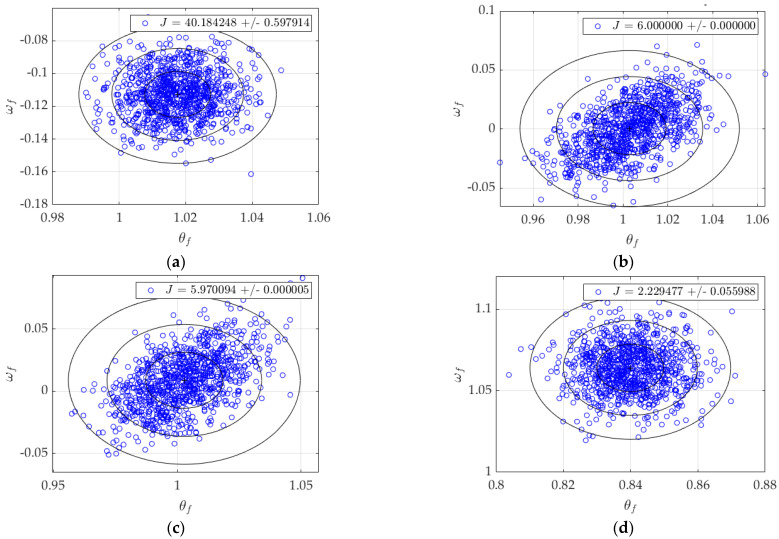
Monte Carlo plots, *N* = 1000. Rings show 1-sigma, 2-sigma, and 3-sigma deviations from the mean, denoted by a +. (**a**) P+V control; (**b**) open-loop optimal control; (**c**) RTOC; (**d**) double-integrator patching filter; (**e**) double integrator, tuned; (**f**) control law inversion patching filter.

**Figure 5 sensors-23-03169-f005:**
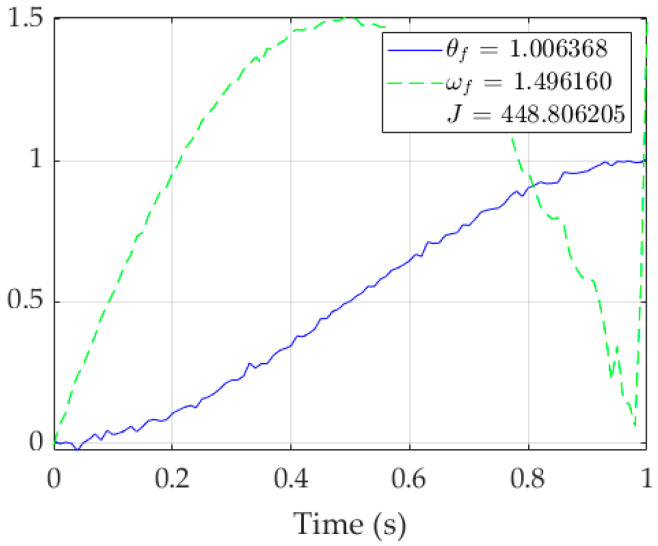
Singular run of RTOC using *pinv*(*A*) without open-loop switching. Note the deterioration in ω(t) starting after time = 0.8 s, indicating the function’s inability to reliably solve the system as it approaches singularity.

**Figure 6 sensors-23-03169-f006:**
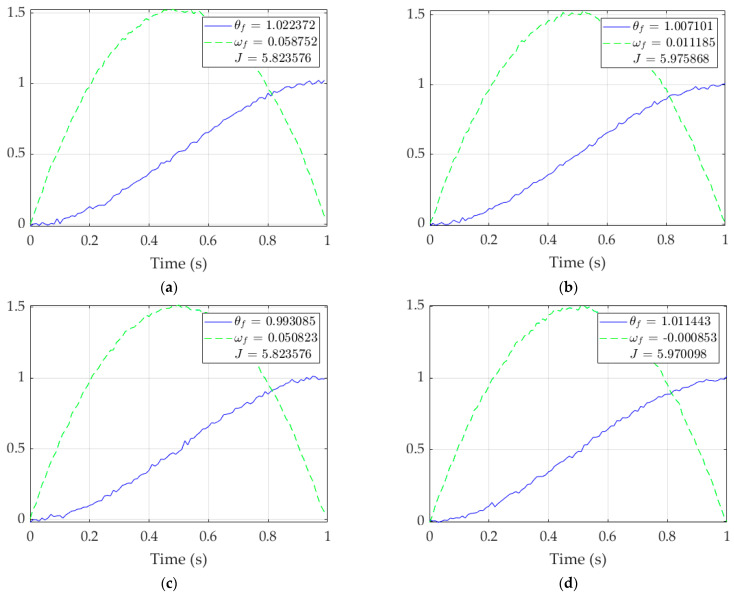
Singular runs of RTOC for each matrix inverse procedure. **Solid blue line**: θ(t); **dashed green line**: ω(t). (**a**) ${\left(A\right)}^{-1}$; (**b**) 1\*A;* (**c**) inv(*A*); (**d**) pinv(*A*). Note the failure of plots (**a**,**c**) to reach the right side, where time = 1 s.

**Table 1 sensors-23-03169-t001:** Numerical results of Monte Carlo simulations, *N* = 1000. σ refers to the standard deviation of a quantity. All other numerical results represent the mean over all simulations.

Architecture	$\theta_{f}$	$\sigma_{\theta }$	$\omega_{f}$	$\sigma_{\omega }$	*J*	$\sigma_{J}$
P+V ^1^	1.0176	0.0099	−0.1128	0.0140	40.1842	0.5979
Open-loop optimal	1.0029	0.0163	0.0004	0.0220	6.0000	0.0000
RTOC	1.0028	0.0225	0.0087	0.0156	5.9701	0.00001
Double Integrator ^1^	0.8399	0.0100	1.0640	0.0146	2.2295	0.0560
Double Int., Tuned ^2^	1.0227	0.0116	−0.0140	0.0510	13.3345	0.7892
Control Inversion ^1^	0.9996	0.0105	−0.0016	0.0155	6.0412	0.1045
P+V ^1^	1.0176	0.0099	−0.1128	0.0140	40.1842	0.5979

^1^ $K_{P}=37,K_{V}=8.5$. ^2^ $K_{P}=78,K_{V}=0.65$.

**Table 2 sensors-23-03169-t002:** Numerical results of Monte Carlo simulations, *N* = 1000. σ refers to the standard deviation of a quantity. All other numerical results represent the mean over all simulations. Percent differences are compared to the benchmark open-loop optimal results.

Architecture	$\theta_{f}$	$\sigma_{\theta }$	$\omega_{f}$	$\sigma_{\omega }$	*Cost*	$\sigma_{J}$
P+V ^1^	1%	−39%	−28,300%	−36%	570%	60%
Open-loop optimal	--	--	--	--	--	--
RTOC	0%	38%	2075%	−29%	0%	0%
Double Integrator ^1^	−16%	−39%	265,900%	−34%	−63%	6%
Double Int., Tuned ^2^	2%	−29%	−3600%	132%	122%	79%
**Control Inversion ^1^**	**0%**	**−36%**	−	**−30%**	**1%**	**10%**

^1^ $K_{P}=37,K_{V}=8.5$. ^2^ $K_{P}=78,K_{V}=0.65$.

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
