# Peer review of "Proposals for Surmounting Sensor Noises"

_sensors, 2023, doi:10.3390/s23063169_

Round 1

Reviewer 1 Report

I think this is a good paper but I would like to highlight some minor things that could make it better.

The introduction does not explain well the reasons for conducting the study and introduces the theoretical framework of the study.

The design of the control system parameters is missing in the test method.

The problem to be solved by robust design is in fact to obtain reliable and competitive technical design with small cost, the parameters of the system need not be too conservative and cannot fail to meet the requirements.

If the variation of the parameters meets the requirements, then the parameters can be selected, and if they do not meet the requirements, then the data obtained by the simulation method can be used to find out which parameters are the most important and which are the least important, and according to this rule, the control system parameters can be designed.

Author Response

  • The introduction does not explain well the reasons for conducting the study and introduces the theoretical framework of the study.
    • This is a fair point. Thanks. Modification of the last paragraph of the intro now puts emphasis on the paper being a comparison of control schemes under the same noise environment.
  • The design of the control system parameters is missing in the test method.
    • The parameter design is now included on lines 114-115 for the method to calculate the two P+V gains. Lines 162-163 indicate the necessary instances to use trial and error.
  • The problem to be solved by robust design is in fact to obtain reliable and competitive technical design with small cost, the parameters of the system need not be too conservative and cannot fail to meet the requirements.
    • Thanks for the comment. We agree, and this rationale motivates mandatory inclusion of Monte Carlo analysis in section 3.
  • If the variation of the parameters meets the requirements, then the parameters can be selected, and if they do not meet the requirements, then the data obtained by the simulation method can be used to find out which parameters are the most important and which are the least important, and according to this rule, the control system parameters can be designed.
    • We agree. In this instance, requirements were not issued, rather a comparison of methods was investigated to advice subsequent implementation.

Reviewer 2 Report

It is not clear what is the objective of the paper. Authors start by introducing a lot of works that deal with the optimization of any kind of sensor (and the only clear relationship among them is their link with spacecraft. Then, they declare a list of novelties that  seems to be invented in that moment without any clear relation with the previous introduction. After that they present some mathematical equations etc that should solve a problem related to signal noise, although it is not clear which is the problem they are solving and for which sensor/s.  With the aim of being effective (and understood by the regular readers of the Sensors Journal I suggest authors to modify the structure of their paper. They should start from an introduction that introduce the problem (and not only the context). If they think it is useful, they could also present a use case specifically dedicated to one or some specific sensors (to let the reader understand the real problem and use it also to explain how the proposal can enhance the situations) Then, they should introduce the state of the art. If the problem is really genetically related to any kind of sensor the state of the art should be related only to the specific works that solve the same issue. Otherwise authors could concentrate only on the specific sensor on which they are focusing their study. Then, a system architecture could help to understand on which part authors are intervening (so if the optimisation is done on the sensor firmware, or on the cloud or on an edge/fog device). Then, it should be easier to present the proposal from a mathematical point of view, but authors should clarify how the formulas are really used. Finally, the tests should be performed in a way that demonstrates how a sensor can really benefit from them.

Author Response

  • It is not clear what is the objective of the paper. Authors start by introducing a lot of works that deal with the optimization of any kind of sensor (and the only clear relationship among them is their link with spacecraft. Then, they declare a list of novelties that seems to be invented in that moment without any clear relation with the previous introduction. After that they present some mathematical equations etc that should solve a problem related to signal noise, although it is not clear which is the problem they are solving and for which sensor/s. With the aim of being effective (and understood by the regular readers of the Sensors Journal I suggest authors modify the structure of their paper.
    1. Thanks for the recommendation. We hope the revised document enhances the clarity of the manuscript’s “flow”.
  • They should start from an introduction that introduce the problem (and not only the context).
    1. Thanks for the recommendation. We hope the revised document enhances the clarity of the manuscript’s “flow”.
  • If they think it is useful, they could also present a use case specifically dedicated to one or some specific sensors (to let the reader understand the real problem and use it also to explain how the proposal can enhance the situations)
    • Great suggestion, thanks. Section 1.1 now includes an overt figure exemplifying the problem to be addressed.
  • Then, they should introduce the state of the art.
    • Great suggestion, thanks. This context has been incorporated into the newly structured section 1.2.
  • If the problem is really genetically related to any kind of sensor the state of the art should be related only to the specific works that solve the same issue. Otherwise authors could concentrate only on the specific sensor on which they are focusing their study.
    • Thanks for the advice. The newly added figure and explicit expression of the problem now include this notion.
  • Then, a system architecture could help to understand on which part authors are intervening (so if the optimization is done on the sensor firmware, or on the cloud or on an edge/fog device).
    • Thanks for the advice. The newly added section 1.1 now includes this notion.
  • Then, it should be easier to present the proposal from a mathematical point of view, but authors should clarify how the formulas are really used.
    • Great suggestion, thanks. The method to use the formulas is now prefaces in a new opening paragraph of section 2 illuding to the detailed implementation in the appendices provided to permit duplication by the readership.
  • Finally, the tests should be performed in a way that demonstrates how a sensor can really benefit from them.
    • Thanks for the suggestion. Hopefully the new revisions will clearly illuminate how the system performance reveals amelioration of the deleterious aspects of the sensors.

Reviewer 3 Report

This work is about the proposals for surmounting sensor noises, which has has certain reference significance for the researchers. There are some specific issues, the authors should address them before acceptance.

1、Compared with other methods for reducing senor nosie, what is the advantage of this work?

2 、Rewrite the abstract to highlight innovation.

3、The specific performance indicators  are not clear in the abstract and conclusion part.

Author Response

  1. Compared with other methods for reducing senor noise, what is the advantage of this work?
  • Thanks for the great recommendation. The abstract has been augmented to present the advantage overtly.
  1. Rewrite the abstract to highlight innovation.
  • Thanks for the great recommendation. The abstract has been augmented to overtly highlight the innovation.
  1. The specific performance indicators are not clear in the abstract and conclusion part.
  • Thanks for the great recommendation. The abstract has been augmented to use the performance indicators as avatars representing the innovation and advantages of the proposed approach.

The manuscript is over 12% larger by word-count representing the revisions made in response to the reviewers’ recommendations.  Thanks to all.

Round 2

Reviewer 2 Report

Although authors added some sentences to address the reviewer comments and the paper itself has been actually improved, unfortunately the paper remains unclear and it is still difficult to understand the problem.

As I'm understanding (now there are some elements to understand something about the problem), the problem is that some sensors are affected by noise that interferes with normal functioning of the sensors themselves.
Therefore, authors are trying to propose methods to improve sensors to deal with the noise and therefore improve the accuracy of the resulting measures.

However:
a) the abstract does not help in understanding such an effort. A reader that does not know anything about the proposal can only understand that there are some noisy sensors that are fused in one device to improve the accuracy etc ... but the accuracy of what? Authors could specify that they are looking exactly at removing the noise through some formulas etc. And they should also declare why they are doing it

b) The state of the art is still the same. The state of the art should be related only to the specific works that solve the same issue. Otherwise, readers cannot understand which is the novelty of the work

c) a system architecture is still needed to help readers understand on which part authors are intervening (so if the optimization is done on the sensor firmware, or on the cloud or on an edge/fog device).

d) The introduced picture Figure 2 creates only more confusion. A use case is a scenario that presents a real "situation" (e.g., sensor X is used to monitor machinery Y and is affected by an error that is recognized in the revealed data ...)

e) Add a sentence to declare that the section offers formulas used in the study do not help in understanding what each formula does and how it is applied to the deletion of noise.

I suggest authors to read other papers published on the same Journal to understand how the paper can be structured.
For instance:
- https://www.mdpi.com/1424-8220/20/3/909
- https://www.mdpi.com/1424-8220/20/3/653

Author Response

  1. a) the abstract does not help in understanding such an effort. A reader that does not know anything about the proposal can only understand that there are some noisy sensors that are fused in one device to improve the accuracy etc ... but the accuracy of what? Authors could specify that they are looking exactly at removing the noise through some formulas etc. And they should also declare why they are doing it

      Thank you for the comments. The abstract has been modified to better establish the main effort of the paper and emphasize the role of different controls to work around the existence of noisy sensors.

  1. b) The state of the art is still the same. The state of the art should be related only to the specific works that solve the same issue. Otherwise, readers cannot understand which is the novelty of the work

      Thank you for the comments. The review of novelties has been edited to outline that the selection of state-of-the-art is related to the issue at hand, overcoming sensor noise in the system rather than within the sensors itself.

  1. c) a system architecture is still needed to help readers understand on which part authors are intervening (so if the optimization is done on the sensor firmware, or on the cloud or on an edge/fog device).

      Thank you for the recommendation. A topology graphic has been added in a new section, 1.3, with an explanation which shows explicitly where we are intervening.

  1. d) The introduced picture Figure 2 creates only more confusion. A use case is a scenario that presents a real "situation" (e.g., sensor X is used to monitor machinery Y and is affected by an error that is recognized in the revealed data ...)

      Thank you for the comments. Additional commentary has been added to Figure 2 and to Section 1.1.

  1. e) Add a sentence to declare that the section offers formulas used in the study do not help in understanding what each formula does and how it is applied to the deletion of noise.

      Thank you for the recommendation. Such a sentence has been added to Section 2.

Round 3

Reviewer 2 Report

At last, authors found their way to explain what they are presenting and the current version definitely

improves the overall presentation of the content.
I don't know if it is still possible, but I would send a further suggestion to the authors: they should still
improve the conclusion section to recap what they did before presenting the results (as already done
in the abstract).

Then, after this minor modification, in my opinion, the paper is ready for publication.

Author Response

We thank the reviewer for the recommended re-structuring and we have completed ceded the request moving the summary of the research effort prior to the summarized results augmenting the Conclusions by division into two sections.